# Sunflower Oil Winterization Using the Cellulose-Based Filtration Aid—Investigation of Oil Quality during Industrial Filtration Probe

**DOI:** 10.3390/foods12122291

**Published:** 2023-06-07

**Authors:** Katarina Nedić Grujin, Tanja Lužaić, Lato Pezo, Branislava Nikolovski, Zoran Maksimović, Ranko Romanić

**Affiliations:** 1Faculty of Technology Novi Sad, University of Novi Sad, Bulevar cara Lazara 1, 21000 Novi Sad, Serbia; katarina.nedic.grujin@gmail.com (K.N.G.); tanja.luzaic@tf.uns.ac.rs (T.L.); barjakb@uns.ac.rs (B.N.); 2Dijamant Ltd., Temišvarski drum 14, 23000 Zrenjanin, Serbia; 3Institute of General and Physical Chemistry, University of Belgrade, Studentski trg 12/V, 11158 Belgrade, Serbia; latopezo@yahoo.co.uk; 4Faculty of Pharmacy, University of Belgrade, Vojvode Stepe 450, 11221 Belgrade, Serbia; zoran_maksimovic@yahoo.com

**Keywords:** sunflower oil, winterization, waxes, cellulose-based filtration aid, optimization

## Abstract

Waxes, phospholipids, free fatty acids, peroxides, aldehydes, soap, trace metals and moisture present in crude sunflower oil have a negative effect on the oil quality and are, therefore, removed during the refining process. Waxes crystallizing at low temperatures are removed during winterization by cooling and filtration. Waxes have poor filtration characteristics and an industrial filtration process must be enhanced by the use of filtration aids, which improve filter cake structure and properties, and consequently prolong the filtration cycle. Today, traditional filtration aids (diatomite, perlite, etc.) being used in the industry are frequently replaced by cellulose-based aids. The aim of this study is to examine the effect of oil filtration assisted by two cellulose-based filtration aids on the chemical parameters (wax, moisture, phospholipids, soaps, and fatty acids), oil transparency, carotenoids, and Fe and Cu content of sunflower oil obtained in an industrial horizontal pressure leaf filter. In order to investigate the mentioned parameters, the following techniques were used: gravimetric (waxes and moisture content), spectrophotometric (phospholipids and carotenoid content and oil transparency), volumetric (soaps and free fatty acids content) as well as inductively coupled plasma mass spectrometry (ICP-MS) for Fe and Cu content. An artificial neural network model (ANN) was employed for the prediction of removal efficiency based on the chemical quality, oil transparency, Fe and Cu content in oils before filtration, as well as filtration aid quantity and filtration time. Cellulose-based filtration aids had multiple beneficial effects; on average, 99.20% of waxes, 74.88% of phospholipids, 100% of soap, 7.99% of carotenoids, 16.39% of Fe and 18.33% of Cu were removed.

## 1. Introduction

Crude sunflower oil contains undesirable components such as phospholipids, free fatty acids, peroxides, aldehydes, trace metals, polymers, waxes, mono- and diacylglycerols, moisture and other volatile compounds [1]. These components are removed during the refining process because of their negative effect on the sensory properties, hydrolytic and oxidative stability of refined oil, as well as oil losses during refining [2,3,4]. The refining process includes pre-refining or degumming, followed by neutralization, decolorization, winterization and deodorization, potentially performed in a different order [3,5,6]. During the refining process, phospholipids and free fatty acids are always removed first, followed by partial removal of heavy metals and pigments [3,7,8,9]. The winterization process primarily removes components that crystallize at low temperatures and causes oil turbidity [10]. In sunflower oil, these components primarily are waxes [11]. The oil waxes present in sunflower oil are mainly long-chain fatty acids and alcohol esters with 34 to 50 C atoms [12]; their melting point is between 70 and 80 °C [13,14]. Most of the waxes originate from sunflower hull [14,15,16]. Only a part of total waxes is removed during the winterization; short-chain waxes containing less than 42 C atoms are present in the final product [12].

The most abundant phospholipids in sunflower oil are phosphatidic acid (69%), phosphatidyl choline (13%), phosphatidyl ethanolamine (13%) and phosphatidyl inositol (5%) [15]. The presence of phospholipids in edible oils inhibits crystallization at low wax concentrations [17]. This primarily relates to phospholipids, free fatty acids and soaps in oils before the winterization process [18]. Free fatty acid content in crude sunflower oil is especially important, directly indicating the oil losses in the form of soap during the neutralization process [6]. The high soap content in the oil, before filtration, leads to a decrease in the porosity of the filtration cake and, during the deodorization, it causes darker oil color and change in the taste [3]. Moisture causes hydrolytic changes, increased free fatty acid content and oil turbidity [19].

Carotenoids are the main pigments in sunflower oil [20,21]. Total carotenoids content is significantly reduced during the refining process, although their presence in oil is convenient due to the antioxidant activity [22,23,24,25]. The oil transparency indicates total pigments content. The total carotenoids content in sunflower oil is inversely proportional to the oil transparency [26].

Filtration of the sunflower oil during the winterization phase is aggravated by the presence of the residual substances. During the filtration, solid particles are separated from fluids, remaining on the filtration medium [27]. Usually, the oil turbidity-causing particles are small and compressible. After a short period of time, these particles cause clogging of the filtration medium pores, reducing filter capacity and filtrate quality. In order to prevent clogging of the filtration medium and to reduce its specific resistance, i.e., to prolong the duration of the filtration cycle, some filtration aid has to be used. This aid should not affect the chemical composition, flavor or taste of the oil and has to be composed of solid, finely porous, incompressible particles. Before the filtration, the filtration aid is applied to the porous partitions of the filter in the form of a flood layer and, during the filtration, it is continuously dosed into the oil before the filter. In this way, the resistance of the filtration layer is significantly reduced and, in addition to the high flow rate, a high degree of clarification is provided [28,29]. Filtration without auxiliaries would not be possible because, if the filter media were used alone, the filters would act as surface filters [30].

Today, filtration aids consisting mainly of natural cellulose fibers are increasingly used [31]. The advantages of cellulose-based filter media are numerous. Low specific weight reduces the consumption of the filtration aid and, thus, reduces the oil loss with the filtration cake. Due to a low density of the fibers, the aid more easily maintains in suspension, without sedimentation. Pre-coat filtration, employing a thin layer of about 0.5 to 1.0 kg m^−^^2^ deposited on the filter medium prior to beginning feed to the filter, is commonly used to protect the filter medium from fouling by trapping solids before they reach the medium and to provide a finer matrix to trap fine solids and assure filtrate clarity as well [32]. Cellulose fibers quickly bridge the openings on the filter, protect it and prevent clogging. Filtration cake can be used in animal feed production because it does not contain toxic substances and silicate crystals. Cellulose fibers do not cause abrasion of filters, pumps and filtration equipment. The filtration cake is easier to remove from the filter due to the fibrous mesh structure [33,34,35].

During the winterization process, between the wax crystallization and the filtration phases, sunflower oil contains other components affecting the quality, color and other oil properties. This paper presents the results of monitoring the operation of an industrial pressure leaf filter, which is a part of the winterization process during the refining of sunflower oil. The filtration process is assisted by the addition of two filtration aids, one for pre-coat and the other for the body-feed application. The focus of the work is on the quality of the filtered oil, depending on the quality of the oil entering the filter and the amount of the added filtration aids. Waxes, moisture, phospholipids, soaps, fatty acids, carotenoids, iron (Fe) and copper (Cu) contents, as well as transparency (at wavelength of 455 nm) of the oil samples were determined. Furthermore, the main objective of this investigation was to explore the potential of forecasting the removal efficiency of mentioned parameters based on data of initial values of chemical parameters, Fe and Cu content, and also the data regarding filtration time and the quantity of filtration aid. The importance of this research is in the introduction of cellulose-based filtration aids with certain technological and ecological advantages compared to classic aids used in the oil refining process. On the other hand, since their price is higher compared to classic filtration aids, it is necessary to prove the effectiveness of these filtration aids so companies would be ready to invest. In this regard, the main objective of this work is to prove the effectiveness of cellulose-based filtration aids and to optimize the process.

## 2. Materials and Methods

### 2.1. Samples

Analyzed samples were taken during the industrial refining of sunflower oil, before and after filtration, as a part of the winterization phase. Industrial processing was carried out during 2019–2020 and the industrial refining capacity was 200 tons of crude sunflower oil per day every 24 h. The obtained oil is produced from sunflower seeds (*Helianthus annuus* L.) grown in the territory of Vojvodina (north of the Republic of Serbia) in 2019.

#### 2.1.1. Industrial Winterization Process

The winterization of sunflower oil takes place after the neutralization phase. The neutral oil was cooled and treated with a dilute sodium hydroxide solution. Then, the oil was crystallized by slow mixing in tanks at 8 °C. After the crystallization, a part of the waxes, along with the soap, were separated at low temperature. Thereafter, the oil was washed and dried. Figure 1 shows the flow chart of the industrial refining of crude sunflower oil with an emphasis on winterization using the cellulose-based filtration aid.

The obtained oil still contained waxes, phospholipids, soap, free fatty acids, moisture and carotenoids. In order to extract the remaining turbidity-causing compounds from the oil, the oil was cooled and filtrated. The filtration was performed on horizontal pressure leaf filter (Amafilter BV, Alkmaar, The Netherlands; surface: 60 m^2^) (Figure 2) at 16 °C. Two filtration aids were used to prepare the filtration suspension—one filtration aid for precoat filtration at the beginning of the process and the other was used for body-feed application, being continuously added during the process.

One commercial cellulose-based aid (FA-P) was used for the application on filters (beige powder, 6% loss on drying, with bulk density of 120–200 g/L; interior mesh aperture: >250 μm (max 5%), >100 μm (5–50%), >32 μm (min 45%)), while the other commercial cellulose-based aid (FA) (light yellow powder, 6% loss on drying, 0.3% oxide ash, with bulk density of 160–220 g/L; interior mesh aperture: >>250 μm (max 1.5%), >>100 μm (max 70%), >>32 μm (max 98%)) was used for the dosing during filtration (JRS, J. Rettenmaier & Sohne GMBH, Rosenberg, Germany).

#### 2.1.2. Sunflower Oil Samples

The oils were sampled before and after horizontal pressure leaf filter. A total of 22 oil filtration cycles (labeled from F_01 to F_22) were monitored. Oils sampled before horizontal pressure leaf filter had different wax content, total carotenoids content, iron and copper content. Moisture content, total phospholipids content, soap and free fatty acids content of examined oil samples before filtration also differed and depended on the efficiency of the previous refining phases (degumming and neutralization). Moreover, 22 oils were sampled after the horizontal pressure leaf filters. About 1 L of oil was sampled in 1 L PET bottles, sealed with original two-part caps and stored in a refrigerator at 4 ± 2 °C before analysis.

### 2.2. Sunflower Oil Quality

Wax content, phospholipid content, soap content, free fatty acid content, moisture content, carotenoid content, iron (Fe), copper (Cu) content and oil transparency were determined in all sunflower oil samples (44 in total).

#### 2.2.1. Waxes Content

The wax content in the oil samples before the horizontal pressure leaf filter was determined by gravimetric method according to Oštrić-Matijašević and Turkulov, 1973 [36]. The key stages in the determination were wax crystallization and “warm” and “cold” extraction. Based on these selective extractions, it was possible to quantitatively determine the wax content, using their characteristics to crystallize and settle in oil at lower temperatures. The wax crystal separation and purification were performed based on their different solubility in different solvents depending on the temperature. The test oil samples were tempered at 6–8 °C to allow the waxes to form crystals. Thereafter, oils were filtered at the same temperature. Wax crystals, as well as part of oil, free fatty acids, and phosphatides remained on the filter paper (MN 619 de, Macherey-Nagel). Purification of the separated wax crystals was performed by “cold” extraction with *n*-hexane at 1–3 °C for 8 h. All the accompanying components were extracted by solvent at mentioned temperature (1–3 °C), so pure wax crystals remained on the filter paper. Later, waxes were extracted by “warm” alcohol for 4–6 h and collected in a laboratory flask. Obtained flask content was cooled to allow the waxes to crystallize and filtered. The filter paper, together with the waxes, was dried at 103 ± 2 °C and measured. Based on the obtained wax mass and initial sample mass, the wax mass fraction was calculated.

The wax content in the oil samples after the horizontal pressure leaf filter was too low, below the limit of quantification by the gravimetric method. In this case, the wax content was practically determined as the threshold of oil turbidity, the smallest amount of wax that causes turbidity in the oil, under the test conditions. The oil was kept for a certain time at a defined low temperature: 0, 5, 7, 12 or 15 °C, observed visually and the time required for oil turbidity to appear was recorded. This method is also used to assess the “resistance” of oil to crystallization and is most often used to control the winterization process [11,37,38]. The oil samples were heated to 130 °C with constant stirring. Later, samples were cooled to 25 °C, placed in an ice bath at 0 °C. After 5.5 h, the oil samples were visually observed for turbidity. If turbidity did not occur, the quantitative (gravimetric) method could not be applied and the wax content was less than 14 mg/kg [11,20,39].

#### 2.2.2. Moisture Content

The moisture content was measured according to ISO 662, 2016 [40].

#### 2.2.3. Total Phospholipids Content

The total phospholipids content was measured spectrophotometrically, using calibration curve, according to AOCS, 1989 [41].

#### 2.2.4. Soap Content

The soap content was determined volumetrically, according to AOCS, 1985 [42]. The oil samples were dissolved in acetone with hydrochloric acid in the presence of the indicator (bromophenol blue).

#### 2.2.5. Free Fatty Acid Content

The free fatty acid content was measured according to ISO 660, 2020 [43].

#### 2.2.6. Total Carotenoids Content

The total carotenoids content (as β–carotene equivalent) was determined by a spectrophotometric method [44] measuring the absorbance of pure oil sample at 445 nm using a UV/VIS spectrophotometer T80+ (“PG Instruments”, Lutterworth, UK).

#### 2.2.7. Oil Transparency

The oil color was determined measuring pure oil sample transparency (% T) at 455 nm using a UV/VIS spectrophotometer T80+ (“PG Instruments”, Lutterworth, UK), as described Dimić and Turkulov, 2000 [45].

#### 2.2.8. Iron and Copper Content

The determination of heavy elements was performed by inductively coupled plasma mass spectrometry (ICP-MS). Approximately 0.5 g of the oil sample was measured and transferred into the Teflon vessels for microwave digestion. Thereafter, 8 mL of concentrated nitric acid (69%, J. T. Baker (Center Valley, PA, USA)) and 1 mL of hydrogen peroxide (30% J. T. Baker (Center Valley, PA, USA)) were added. Microwave digestion system (Ethos One, Milestone, Italy) was used for the digestion of the oil samples. A mixture of HNO_3_/H_2_O_2_ was provided as a blank sample to assess the contamination. The samples were digested according to the manufacturer’s recommendation (Milestone Ethos Microwave Digestion System method). After cooling at room temperature, the digests were diluted into a 25 mL plastic flask and transferred to vessel for further analysis. Quantification of Fe and Cu was conducted using acidified aqueous metal standards (J. T. Baker, Center Valley, PA, USA) by an external calibration procedure.

### 2.3. Machine Learning Model

Various classical machine learning models are widely utilized in modeling across different scientific fields. These models include artificial neural network (ANN), random forest regression (RFR), support vector machine (SVM), extreme learning machine (ELM), K-nearest neighbors (KNN) and decision tree (DT). SVM, a discriminant technique based on statistical learning theory, is recognized for its exceptional generalization ability. By striking a balance between model complexity and training error, the optimal network is achieved [46]. ELM constructs a single-layer feed-forward network by randomly generating input weights and biases for the hidden layers [47].

State-of-the-art machine learning techniques offer a diverse range of options for sequence data, such as ensemble learning models, such as XGBoost [48], LightGBM [49] and CatBoost. XGBoost stands out for its high prediction accuracy and interpretability. LightGBM allows efficient handling of large datasets and GPU training. Compared to XGBoost, LightGBM models have demonstrated superior accuracy and faster performance. Data fusion, incorporating gradient boosting with categorical attributes supported by the CatBoost algorithm, enhances forecasting accuracy [50].

In this paper, artificial neural network model [51,52], as a well-known and broadly accepted machine learning technique, was utilized to contemplate the removal efficiency of chemical parameters (the contents of wax, moisture, phospholipids, soap, fatty acids and carotenoids), oil transparency (at wavelength of 455 nm), Fe and Cu content after filtration process, based on data of initial values of chemical parameters, Fe and Cu content, and, also, the data regarding filtration time and the quantity of filtration aid.

The ANN modeling technique was chosen for prediction purposes due to its proven efficiency in approximating nonlinear functions [53,54].

The ANN model building structure was based on the multi-layer perceptron model (MLP) scheme, comprised of three layers (input, hidden and output).

The MLP-formed ANN model could be presented using matrix notation, with weight and bias coefficients associated to the hidden and output layer written in matrices *W*_1_, *B*_1_, *W*_2_ and *B*_2_, with *Y* as the output variables matrix, *f*_1_ and *f*_2_ as activation functions in the hidden and output layers, and with *X* as the matrix of input variables [55]:(1)$$Y=f_{1}\left(W_{2}\cdot f_{2}\left(W_{1}\cdot X+B_{1}\right)+B_{2}\right)$$

Before the calculation, the experimentally obtained database consisting of measured input and output parameters was transformed using min−max normalization scheme. This database was randomly divided into training, cross-validation, and testing groups (60%, 20% and 20%, respectively). Throughout the learning procedure, ANN inputs were supplied with a training set of parameters in order to establish the optimal number of neurons in the hidden layer to estimate the weights and bias coefficients and non-linear activation functions for every neuron in the ANN model.

During the iterative process of calculating weights and biases coefficients and testing different activation functions for the hidden and output layers, the Broyden−Fletcher−Goldfarb−Shanno (BFGS) algorithm was employed. Various activation functions were explored, including hyperbolic tangent, logistic sigmoidal, exponential and identity functions. The identity function directly passes the activation level from the input as the output of the neurons. Logistic sigmoid uses the S-shaped logistic sigmoid function, producing output values in the range of 0 to +1. The hyperbolic tangent function (tanh) is another symmetric S-shaped (sigmoid) function, with output values ranging from −1 to +1. It often outperforms the logistic sigmoid function due to its symmetry. The exponential function utilizes the negative exponential activation function [56]. A sequence of distinct MLP-formed ANN layouts was investigated, altering the number of hidden neurons (between 5 and 20) introducing random initial values of weights and biases coefficients. The learning procedure of the network was repeated 100,000 times [57]. The optimization setup included the minimization of the square error. It is assumed that the successful training was reached when learning and cross-validation curves approached zero.

#### Global Sensitivity Analysis

Yoon’s interpretation method was used to determine the relative influence of input variables on the removal efficiency of chemical parameters (the contents of wax, moisture, phospholipids, soap, free fatty acids and carotenoids), oil transparency (at wavelength of 455 nm) and Fe and Cu content after filtration process [58]. This method was applied on the basis of the weight coefficients of the developed ANN.

### 2.4. Descriptive Statistics

The results were revealed by mean value ± standard deviation (n = 3). Data were analyzed by one-way analysis of variance (ANOVA) with a Tukey HSD test. It was used to determine significant differences at the significance level *p* < 0.05. Statistical processing of the obtained results and ANN modeling was performed using Statistica version 13.5.0.17 (StatSoft, Tulsa, OK, USA).

## 3. Results and Discussion

The investigation results of the content of waxes, moisture, phospholipids, soap, free fatty acids, iron, copper, total carotenoids content, as well as oil transparency before and after filtration are shown in Table 1 and Table 2. The content of the mentioned parameters in the tested oil samples depends on the crude oil itself and on the previous stages of refining, while the content after filtration is affected by the filtration conditions (quantity of added filtration aid and filtration time).

### 3.1. Artificial Neural Network Model

To investigate the removal efficiency of chemical parameters during winterization, by cooling and filtration, an artificial neural network (ANN) technique was employed. The structure and outcomes of the ANN model depend on the initial assumptions of matrix parameters, which are crucial for building and fitting the ANN to experimental data. Moreover, the behavior of the ANN model can be influenced by the number of neurons in the hidden layer. To address this concern, each network topology was iterated 100,000 times to minimize random correlations caused by initial assumptions and random weight initialization. Through this approach, the highest r^2^ value during the training cycle was achieved when using nine hidden neurons for constructing the ANN model (Figure 3a).

The model underwent training for 100 epochs and Figure 3b displays the training results, specifically the train accuracy and error (loss). During the training process, the training accuracy consistently improved as the number of training cycles increased, up until the 70th to 80th epoch. At this point, the training accuracy reached a nearly constant value. The 70th to 80th epoch yielded the highest train accuracy and lowest train loss. However, after this point, a slight increase in train accuracy and decrease in train loss were observed, indicating the onset of overfitting. Going beyond 80 epochs for training could potentially lead to significant overfitting, while training for 70 epochs would be sufficient to achieve high model accuracy without risking overfitting (Figure 3b).

The developed optimal neural network model showed the adequate generalization capabilities for the prediction of the removal efficiency of chemical parameters (the contents of wax, moisture, phospholipids, soap, fatty acids and carotenoids), oil transparency (at wavelength of 455 nm) and Fe and Cu content after filtration process (Table 3), compared to initial parameters, based on data of initial values of chemical parameters, Fe and Cu content, and also the data regarding filtration time and the quantity of filtration aid. The optimum number of neurons in the hidden layer of ANN model was nine (network MLP 2-9-9); while the r^2^ values were equal to 1.000, during the training cycle r^2^ for output variables, hidden and output layer activation functions were logistic sigmoid (Table 4).

The developed ANN model consisted of 117 weights−bias coefficients due showing the high nonlinearity of the system [59,60].

For the ANN model, the model-calculated values were not too close to the experimental values in most cases, in terms of r^2^ values, while the sum of squares (SOS) values acquired using the ANN model were of the same order of magnitude as experimental errors for outputs mentioned in the literature [55,56].

### 3.2. Wax Content

The wax content is the main refined oil quality and winterization efficiency indicator [61]. In the oil samples before the horizontal pressure leaf filter, the wax content was already partially reduced, as a consequence of neutralization and wax separation, but not enough to meet the quality of less than 14 mg kg^−1^ waxes in refined oil [3]. In our experiment, different wax content was found in the analyzed oils before the horizontal pressure leaf filter. The highest wax content (W_in) was determined in the oil sample F_08 and amounted to 549 ± 11 mg kg^−1^, while the lowest wax content was sample F_16 (281 ± 8 mg kg^−1^). According to the results shown in Table 2, waxes content in sunflower oils after filtration (W_out) using cellulose-based filtration aids was very low. Waxes were practically completely removed from sunflower oil in all oil samples (98.97–99.42%, compared to oil samples before filtration). The highest wax content after horizontal pressure leaf filters was noticed in samples F_08 and F_13 (3.21 ± 0.06 1 and 3.21 ± 0.05 mg kg^−1^, respectively), while the lowest wax content values were 2.89 ± 0.04 and 2.89 ± 0.05 mg kg^−1^, found in F_09 and F_16 samples. Similar results were reported by Mitrović et al., 2009 [62], namely, the wax content before filtration was 590 and 700 mg/kg and after filtration <14 mg kg^−1^. The wax content in the winterized oil obtained by the turbidimetric method ranged from 5.91 mg kg^−1^ to 16.07 mg kg^−1^ [63]. Correlation investigation found that the wax removal efficiency on the horizontal pressure leaf filters assisted by the cellulose-based aids was higher, as the wax content in the oils before filtration was higher (R = 0.98; *p* = 0.00). The same conclusion applied to global sensitivity analysis. Namely, wax content in the initial samples was the most important parameter for the prediction of W_out, with relative importance of +33.40% (Figure 4a). The negative influence on W_out was observed for Cu_in, which showed the relative importance of −14.95%. Total carotenoids (−9.69%) and moisture (−7.81%) content in the initial oil negatively influenced the wax removal.

### 3.3. Total Phospholipids and Soap Content

Phospholipids and free fatty acids are present as accompanying minor components of crude sunflower oil and they have to be removed during the oil refining process [64]. Most of the phospholipids and soap were removed during the degumming and neutralization phase. The accompanying minor components play an extremely important role in wax removal during winterization phase [65]; consequently, total phospholipids and soap content in the oils before and after filtration assisted by the cellulose-based aids were examined. The highest total phospholipids content in the oils before filtration was noticed in samples F_08 and F_13 and amounted to 80 ± 5 mg/kg and 80 ± 6 mg/kg, respectively, while the lowest total phospholipids content (5 ± 1 mg/kg) was found in sample F_15 (Table 1). In the oil sample F_05, before filtration, phospholipids were not detected. Similar results were reported in enzymatic degummed oil (between 63.5 and 65.25 mg/kg) [66], while significantly higher total phospholipid content values in degummed sunflower oil (between 470 and 1230 mg/kg) were found by Lamas et al., 2016 [64]. The oil samples after the horizontal pressure leaf filter (F_05, F_06, F_07, F_12, F_14, F_15, F_18 and F_22) did not contain phospholipids, while, in the rest of the samples, the presence of total phospholipids (from 4 ± 2 to 38 ± 4 mg/kg) was detected, as shown in Table 2. The phospholipids reduction was in the range of 46.67% (F_17) to as much as 100.00% (phospholipids were not detected after filtration). The filtration time had the highest negative influence on phospholipids removal (−18.88%), as well as soap content (−17.11%), while Fe (+10.25%), total carotenoids (8.77%) and Cu (+6.23%) content positively influenced the phospholipids reduction (Figure 4c).

During the classical alkaline refining, free fatty acids (FFA) using a sodium hydroxide (NaOH) were removed in the form of soap. Obtained soaps are water soluble, so oil washing reduces soap content easily to minimum values, below 50 mg/kg [67], even below 10 mg/kg. The soap content in sunflower oil changes in certain refining phases, including winterization. In addition to waxes and phospholipids, the soap content also differed in the oils before and after filtration. In all examined oils sampled before filtration (S_in), the presence of soap was detected. The highest soap content was found in oil sample F_02 (119 ± 3 mg/kg) and the lowest value 30 ± 4 mg/kg was noticed in F_15 sample (Table 1). However, sunflower oils sampled after filtration had very low soap content; their presence was not detected by the applied method, i.e., soap removal efficiency was 100 ± 0.00% (Table 3).

### 3.4. Moisture Content and Free Fatty Acids Content

The moisture content is an important quality indicator of crude and refined oils. The presence of moisture in the oil is undesirable for several reasons, primarily economic interest and quality issues [68]. The moisture content in oils sampled before filtration (M_in) ranged from 0.06 ± 0.01% (F_22) to 0.23% (F_6 and F_11) (Table 1). After filtration, the moisture content values vary in the range from 0.08 ± 0.01% (F_18) to 0.28 ± 0.01% (F_04).

Free fatty acids are formed by hydrolytic cleavage of triacylglycerol during inadequate storage and preparation of seeds for pressing and during pressing [69]. The resulting free fatty acids are very susceptible to oxidation, exhibit pro-oxidative action, affect sustainability and oxidative stability of the oil [70]. Free fatty acid content in oils sampled before filtration (FFA_in) ranged from 0.07 ± 0.01% (F_19) to 0.12 ± 0.01% (F_15). Oils after filtration contained from 0.08 ± 0.01% to 0.11 ± 0.01% free fatty acids. The obtained moisture and FFA content values in the oils after the filter in relation to the oils before filtration generally indicate a decrease in these parameters. However, in some oil samples (with very low moisture and FFA content before filtration) an increase in these parameters was noticed as a consequence of the error of the method due to very low values, so actual increase did not exist. Global sensitivity analysis showed that the moisture content in the oil before filtration, filtration time and phospholipid content had the highest positive influence on the moisture removal in oils after filtration, +22.06%, +20.96% and +17.28%, respectively, while FFA (+20.95%) and wax (13.36%) content in the oil before filtration positively influenced the FFA removal (Figure 4b,d).

### 3.5. Total Carotenoids Content and Oil Transparency

Carotenoids are a large group of polyunsaturated hydrocarbons, composed of isoprene residues [71] and dominant pigments of sunflower oil [20,21]. The oil filtration assisted by cellulose filtration aid led to a reduction in total carotenoids content in oils. TTC in the oil samples before filtration ranged from 6.55 ± 0.04 (sample F_08) to 4.75 ± 0.01 mg/kg (F_05) (Table 1), while, in oils after filtration, lower values were noticed (from 5.98 ± 0.02 mg/kg–F_08 to 4.57 ± 0.05 mg/kg–F_05) (Table 2). The average efficiency of total carotenoids removal was 7.99 ± 3.20% (Table 3).

The oil transparency values are a consequence of different pigment content, primarily total carotenoids. The total pigments content in the oil is inversely proportional to the oil transparency value [72]. In the oils before filtration, the oil transparency values varied from 58.6 ± 0.1% (F_02) to 47.2 ± 0.1% (F_18). Due to the total carotenoids removal, an increase in the value of transparency was also noticed and the obtained values ranged from 50.0 ± 0.1% (F_18) to 61.7 ± 0.0% (F_02). The oil transparency values of oils sampled after filtering increased on average by 6.01 ± 2.46%. Global sensitivity analysis (Figure 4e,f) showed that Cu and soap content had the highest positive influence on total carotenoids removal, +20.53% and +15.25%, respectively, and the highest negative influence on oil transparency (−10.18% and 21.51%). Moisture and phospholipid content in oils before filtration negatively influenced the total carotenoids reduction.

### 3.6. Iron and Copper Content

Traces of metal ions increase the oxidation degree of edible oils and fats, and thus affect the sensory characteristics of oils [73,74]. The obtained values of iron content in the tested samples ranged from 0.583 ± 0.015 (F_05) to 2.670 ± 0.220 (F_01), while the copper content ranged from 0.016 ± 0.003 (F_08) to 0.051 ± 0.006 (F_02) (Table 1). Similar results were reported previously; Lamas et al., 2016 [64] found 2.260 ± 0.170 and 4.150 ± 0.160 mg/kg of iron in degummed sunflower oil, while slightly lower values of iron (1.02 ± 0.10 and 3.05 ± 0.23 mg/kg) and copper content of 0.62 ± 0.06 and 1.32 ± 0.11 mg/kg were reported by Lamas et al., 2014 [66]. Maximum values of iron (1.009 ± 0.006 mg/kg) and copper (0.045 ± 0.001 mg/kg) content in the oils sampled after filtration were obtained in F_08 and F_15 samples, respectively. Obtained iron and copper content values in oils sampled after horizontal pressure leaf filter are in accordance with the values prescribed by the Codex Alimentarius Standard [67]. The efficiency of iron removal was 16.39%, while copper was removed, on average, 18.33%. Total phospholipid content in the oils before filtration positively influenced on Fe (+18.23%) and Cu (+5.72%) removal, while waxes in the oil before filtration mostly hindered the Fe and Cu reduction, −21.96% and −20.46%, respectively (Figure 4g,e).

## 4. Conclusions

Winterization process and oil filtration assisted by cellulose-based filtration aids were proven to be very effective in wax removal; namely, over 99% waxes were removed. According to the obtained results, filtration with filtration aids based on cellulose had multiple beneficial effects during filtration of sunflower oil; in addition to the wax content, the total phospholipid content (50–100%) and the soap content (100%) were also reduced.

The results of this study disclose the removal efficiency of chemical parameters (the contents of wax, moisture, phospholipids, soap, fatty acid and carotenoids), oil transparency and Fe and Cu content after filtration process, based on data of initial values of chemical parameters, oil transparency, Fe and Cu content, and also the data regarding filtration time and the quantity of filtration agent. The artificial neural network model was adequate for the prediction of output variables (the r^2^ values during training cycle for these variables were 1.000). Results of the global sensitivity analysis showed that the quantity of tested cellulose filtration aids and the filtration time accelerated the removal of only some of the tested parameters, which opens the possibility of further testing of the combination of these aids. The experiments had been planned in a way that the waxes should have been removed from the oil almost completely, and that goal was achieved. Based on the obtained results, it was concluded that there is a possibility to change the quantity of added filtration aid and to achieve the desired result. Therefore, future plans are to carry out research on the optimization of the filtration aid quantity and filtration time, aiming to reduce the operational costs.

## Figures and Tables

**Figure 1 foods-12-02291-f001:**
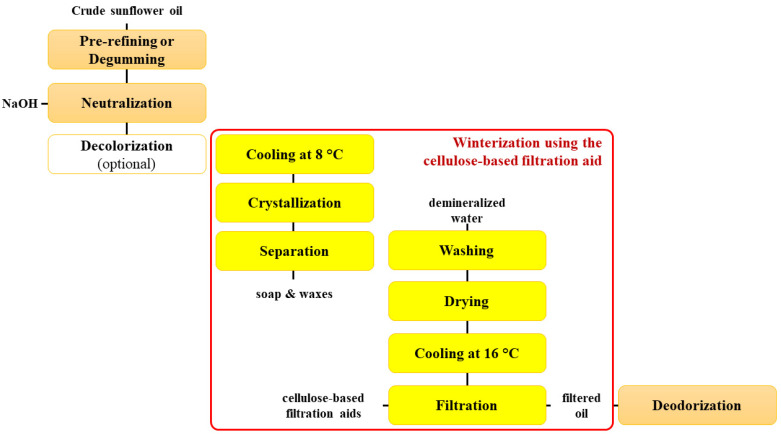
Flow chart of the industrial refining of crude sunflower oil.

**Figure 2 foods-12-02291-f002:**
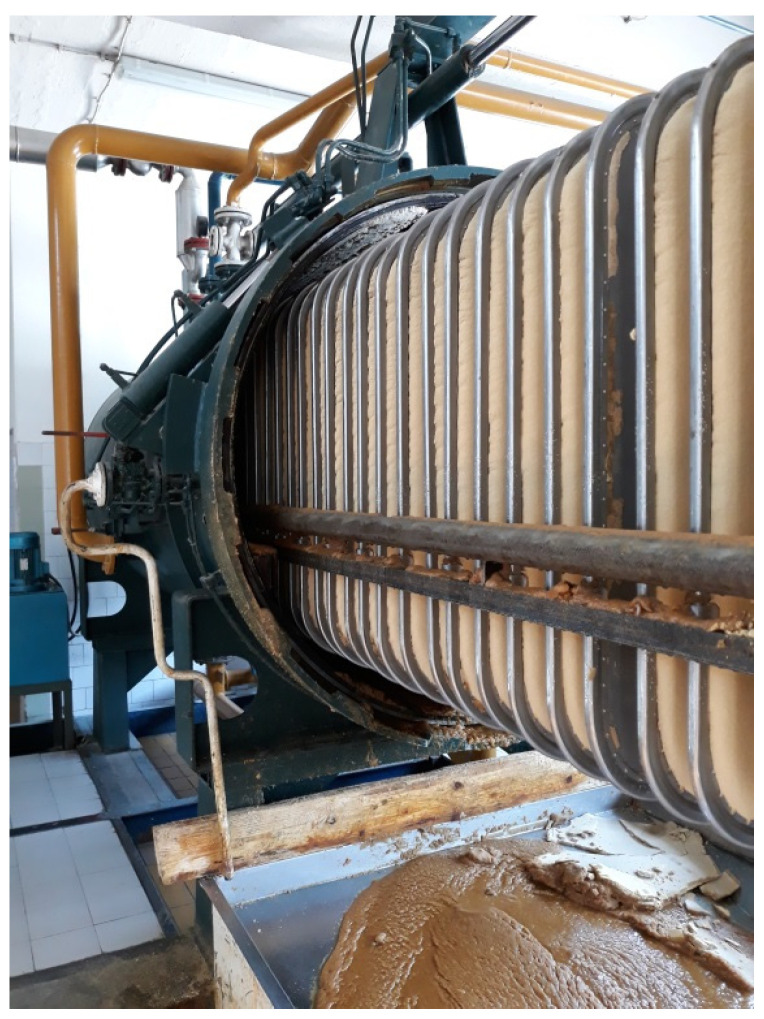
Horizontal pressure leaf filter (Amafilter BV, Alkmaar, The Netherlands; surface: 60 m^2^) used in the industrial winterization process.

**Figure 3 foods-12-02291-f003:**
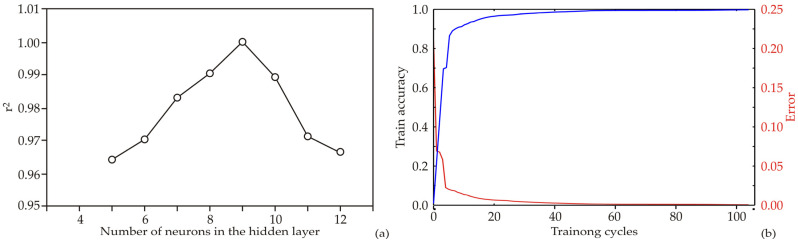
ANN calculation: (**a**) the dependence of the r^2^ value of the number of neurons in the hidden layer in the ANN model; (**b**) training results per epoch.

**Figure 4 foods-12-02291-f004:**
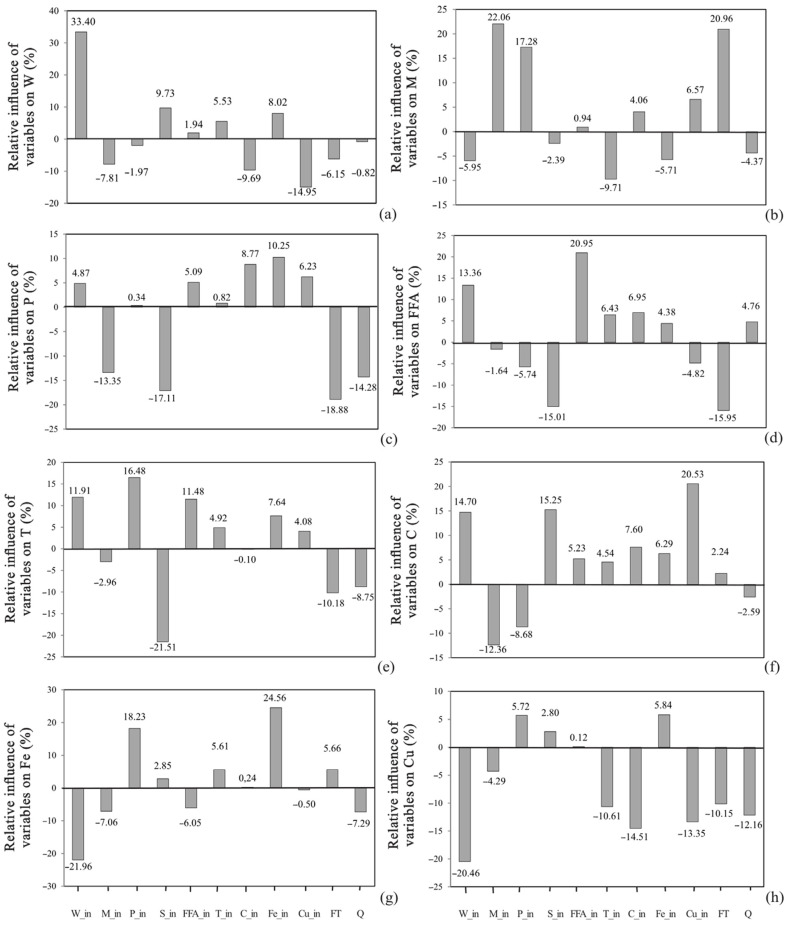
Relative influence (viewed from left to right) of waxes (W_in), moisture (M_in), phospholipids (P_in), soap (S_in), free fatty acids (FFA_in), oil transparency (T_in), total carotenoids content (C_in), iron (Fe_in), copper (Cu_in), filtration time (FT) and the quantity of filtration agent (Q) in oil samples before filtration on the waxes (**a**), moisture (**b**), phospholipids (**c**), free fatty acids (**d**), oil transparency (**e**), total carotenoids content (**f**), iron (**g**), copper and (**h**) removal efficiency.

**Table 1 foods-12-02291-t001:** Results of the content of waxes (W_in), moisture (M_in), free fatty acids (FFA_in), phospholipids (P_in), soap (S_in), total carotenoids content (C_in), iron (Fe_in), copper (Cu_in) and oil transparency (T_in) in oil samples before filtration, as well as the quantity of filtration aid (Q) and filtration time (FT).

Filtration Cycle	Parameter										
	W_in (mg kg^−1^)	M_in (%)	FFA_in (%)	P_in (mg kg^−1^)	S_in (mg kg^−1^)	C_in (mg kg^−1^)	Fe_in (mg kg^−1^)	Cu_in (mg kg^−1^)	T_in (%)	FT (h)	Q (kg)
F_01	382 ± 8 ^e,f^	0.20 ± 0.01 ^c,d,e,f,g^	0.10 ± 0.02 ^b,c,d,e^	45 ± 3 ^e,f^	105 ± 5 ^j,k^	5.38 ± 0.01 ^d,e^	2.67 ± 0.22 ^j^	0.03 ± 0.00 ^a,b,c^	54.8 ± 0.1 ^i^	17	455
F_02	366 ± 7 ^d,e^	0.18 ± 0.01 ^b,c,d,e,f^	0.11 ± 0.01 ^d,e^	12 ± 2 ^a,b^	119 ± 3 ^l^	4.83 ± 0.03 ^a,b^	0.79 ± 0.08 ^b,c,d,e,f^	0.05 ± 0.01 ^e^	58.6 ± 0.1 ^l^	17	430
F_03	509 ± 9 ^j,k^	0.19 ± 0.02 ^b,c,d,e,f,g^	0.08 ± 0.02 ^a,b^	72 ± 3 ^h,i,j^	72 ± 3 ^f,g^	5.13 ± 0.02 ^c^	0.88 ± 0.02 ^c,d,e,f,g^	0.03 ± 0.00 ^a,b,c^	56.4 ± 0.1 ^j,k^	13	405
F_04	547 ± 9 ^l^	0.12 ± 0.01 ^a,b,c,d^	0.08 ± 0.00 ^a,b,c^	79 ± 1 ^j^	109 ± 3 ^k^	4.82 ± 0.01 ^a,b^	0.99 ± 0.09 ^g,h^	0.03 ± 0.00 ^a,b^	58.2 ± 0.1 ^l^	11	330
F_05	403 ± 8 ^f^	0.21 ± 0.01 ^d,e,f,g^	0.10 ± 0.01 ^c,d,e^	nd	56 ± 4 ^e^	4.75 ± 0.01 ^a^	0.58 ± 0.01 ^a^	0.03 ± 0.00 ^a,b,c,d^	56.2 ± 0.1 ^j^	16	455
F_06	326 ± 7 ^b^	0.23 ± 0.01 ^e,f,g^	0.09 ± 0.01 ^b,c,d,e^	17 ± 2 ^b^	54 ± 3 ^d,e^	6.53 ± 0.04 ^k^	0.76 ± 0.03 ^b,c,d,e,f^	0.05 ± 0.01 ^c,d,e^	47.4 ± 0.1 ^a,b^	29	455
F_07	295 ± 7 ^a^	0.15 ± 0.01 ^a,b,c,d,e^	0.11 ± 0.01 ^d,e^	31 ± 2 ^c^	49 ± 2 ^c,d,e^	6.43 ± 0.02 ^j^	0.72 ± 0.03 ^a,b,c,d,e^	0.05 ± 0.00 ^b,c,d,e^	48.1 ± 0.1 ^c^	31	450
F_08	549 ± 11 ^l^	0.11 ± 0.01 ^a,b,c^	0.10 ± 0.01 ^c,d,e^	80 ± 5 ^j^	87 ± 4 ^h,i^	6.55 ± 0.04 ^k^	1.13 ± 0.11 ^i^	0.02 ± 0.00 ^a^	47.7 ± 0.1 ^b,c^	16	455
F_09	288 ± 7 ^a^	0.19 ± 0.07 ^e,f,g^	0.10 ± 0.01 ^d,e^	41 ± 6 ^d,e^	40 ± 3 ^b,c^	5.43 ± 0.01 ^e,f^	0.65 ± 0.02 ^a,b^	0.44 ± 0.02 ^f,g^	54.0 ± 0.1 ^h^	28	455
F_10	526 ± 6 ^k,l^	0.20 ± 0.01 ^f,g^	0.10 ± 0.01 ^d,e^	76 ± 8 ^i,j^	100 ± 3 ^j^	5.30 ± 0.01 ^d^	0.73 ± 0.03 ^b,c,d,e,f^	0.42 ± 0.01 ^f^	56.0 ± 0.1 ^k^	11	355
F_11	355 ± 8 ^c,d^	0.23 ± 0.08 ^f,g^	0.11 ± 0.00 ^b,c,d,e^	40 ± 9 ^d,e^	50 ± 6 ^d,e^	5.88 ± 0.01 ^h^	0.85 ± 0.05 ^c,d,e,f,g^	0.44 ± 0.01 ^g^	52.0 ± 0.1 ^f^	20	455
F_12	490 ± 12 ^i,j^	0.21 ± 0.02 ^g^	0.10 ± 0.01 ^d,e^	60 ± 1 ^g^	75 ± 11 ^f,g^	5.80 ± 0.01 ^g^	0.73 ± 0.03 ^b,c,d,e,f^	0.04 ± 0.00 ^b,c,d,e^	53.0 ± 0.1 ^g^	10	285
F_13	520 ± 10 ^k^	0.19 ± 0.03 ^e,f,g^	0.10 ± 0.01 ^d,e^	80 ± 6 ^j^	90 ± 12 ^i^	5.47 ± 0.01 ^f^	0.64 ± 0.02 ^a,b^	0.04 ± 0.00 ^b,c,d,e^	55.0 ± 0.1 ^i^	7	250
F_14	494 ± 8 ^j^	0.19 ± 0.05 ^f,g^	0.09 ± 0.01 ^e^	79 ± 4 ^i,j^	75 ± 6 ^f,g^	5.54 ± 0.05 ^g^	2.67 ± 0.22 ^a,b,c,d,e^	0.04 ± 0.00 ^b,c,d,e^	51.0 ± 0.1 ^e^	9	310
F_15	287 ± 4 ^a^	0.15 ± 0.02 ^a,b,c,d,e^	0.12 ± 0.01 ^e^	5 ± 1 ^a^	30 ± 4 ^a^	5.70 ± 0.01 ^g^	0.79 ± 0.08 ^e,f,g^	0.05 ± 0.00 ^c,d,e^	53.0 ± 0.1 ^g^	37	355
F_16	281 ± 8 ^a^	0.12 ± 0.04 ^a,b,c,d^	0.09 ± 0.01 ^a,b,c,d^	50 ± 12 ^f^	33 ± 6 ^a,b^	5.34 ± 0.01 ^d^	0.71 ± 0.01 ^f,g^	0.05 ± 0.00 ^d,e^	51.2 ± 0.1 ^e^	39	455
F_17	340 ± 6 ^b,c^	0.20 ± 0.05 ^c,d,e,f,g^	0.09 ± 0.01 ^a,b,c,d^	60 ± 7 ^g^	40 ± 5 ^b,c^	5.87 ± 0.01 ^h^	0.91 ± 0.03 ^c,d,e,f,g^	0.04 ± 0.00 ^b,c,d,e^	49.8 ± 0.1 ^d^	27	455
F_18	351 ± 9 ^b,c,d^	0.08 ± 0.03 ^a^	0.11 ± 0.02 ^d,e^	36 ± 4 ^c,d^	45 ± 4 ^c,d^	6.17 ± 0.01 ^i^	0.92 ± 0.03 ^d,e,f,g^	0.04 ± 0.00 ^b,c,d,e^	47.0 ± 0.1 ^a^	30	450
F_19	443 ± 5 ^g,h^	0.09 ± 0.02 ^a,b^	0.07 ± 0.01 ^a^	65 ± 2 ^g,h^	70 ± 3 ^f^	5.40 ± 0.01 ^e^	0.88 ± 0.02 ^c,d,e,f,g^	0.04 ± 0.00 ^b,c,d,e^	53.0 ± 0.1 ^g^	13	375
F_20	466 ± 10 ^h,i^	0.18 ± 0.05 ^b,c,d,e,f,g^	0.08 ± 0.01 ^a,b,c^	75 ± 8 ^i,j^	100 ± 6 ^j^	5.30 ± 0.01 ^d^	0.89 ± 0.03 ^a,b,c^	0.04 ± 0.01 ^b,c,d,e^	53.0 ± 0.1 ^g^	12	375
F_21	441 ± 9 ^g^	0.08 ± 0.01 ^a^	0.08 ± 0.01 ^a,b^	70 ± 1 ^h,i^	80 ± 2 ^g,h^	5.20 ± 0.01 ^cd^	0.87 ± 0.02 ^a,b,c,d^	0.05 ± 0.00 ^c,d,e^	54.0 ± 0.1 ^h^	14	405
F_22	328 ± 7 ^b^	0.06 ± 0.01 ^a^	0.09 ± 0.01 ^a,b,c,d^	10 ± 3 ^a,b^	45 ± 4 ^c,d^	4.90 ± 0.01 ^b^	0.68 ± 0.02 ^c,d,e,f,g^	0.04 ± 0.00 ^b,c,d,e^	55.0 ± 0.1 ^i^	21	435

nd—not detected. Different lower-case letters in the same column indicate significantly different values (*p* ˂ 0.05), according to post hoc Tukey’s HSD test.

**Table 2 foods-12-02291-t002:** Results of the content of waxes (W_out), moisture (M_out), free fatty acids (FFA_out), phospholipids (P_out), soap (S_out), total carotenoids content (C_out), iron (Fe_out), copper (Cu_out), as well as oil transparency (T_out) in oil samples after filtration.

Filtration Cycle	Parameter								
	W_out (mg kg^−1^)	M_out (%)	FFA_out (%)	P_out (mg kg^−1^)	S_out (mg kg^−1^)	C_out (mg kg^−1^)	Fe_out (mg kg^−1^)	Cu_out (mg kg^−1^)	T_out (%)
F_01	2.94 ± 0.05 ^a,b,c^	0.18 ± 0.01 ^d,e,f,g,h^	0.10 ± 0.01 ^a,b,c,d^	21 ± 1 ^c,d^	nd	5.02 ± 0.04 ^d,e^	0.93 ± 0.02 ^a^	0.02 ± 0.00 ^a,b^	58.6 ± 0.0 ^h,i^
F_02	3.01 ± 0.08 ^a,b,c,d,e,f^	0.21 ± 0.01 ^g,h^	0.10 ± 0.01 ^a,b,c,d^	4 ± 2 ^a^	nd	4.60 ± 0.03 ^a^	0.56 ± 0.02 ^f,g^	0.04 ± 0.00 ^f,g,h,i^	61.7 ± 0.0 ^l^
F_03	3.19 ± 0.08 ^e,f^	0.19 ± 0.01 ^e,f,g,h^	0.08 ± 0.01 ^a,b^	12 ± 3 ^b^	nd	4.83 ± 0.06 ^c^	0.75 ± 0.03 ^d,e,f^	0.03 ± 0.00 ^c,d,e^	58.6 ± 0.5 ^h,i^
F_04	3.16 ± 0.07 ^d,e,f^	0.28 ± 0.01 ^i^	0.08 ± 0.01 ^a^	38 ± 4 ^f^	nd	4.70 ± 0.01 ^b^	0.64 ± 0.03 ^h^	0.03 ± 0.00 ^b,c,d^	59.6 ± 0.1 ^j,k^
F_05	3.00 ± 0.13 ^a,b,c,d,e,f^	0.27 ± 0.01 ^i^	0.08 ± 0.01 ^a,b^	nd	nd	4.57 ± 0.05 ^a^	1.01 ± 0.01 ^b^	0.02 ± 0.00 ^a,b,c^	59.0 ± 0.1 ^i,j^
F_06	2.95 ± 0.07 ^a,b,c,d^	0.11 ± 0.01 ^a,b,c^	0.08 ± 0.01 ^a,b^	nd	nd	5.51 ± 0.04 ^h^	0.52 ± 0.01 ^f,g^	0.04 ± 0.01 ^g,h,i^	53.7 ± 0.1 ^c,d^
F_07	2.95 ± 0.07 ^a,b,c,d^	0.12 ± 0.01 ^a,b,c,d^	0.09 ± 0.01 ^a,b,c^	nd	nd	5.72 ± 0.01 ^j^	0.71 ± 0.01 ^c^	0.04 ± 0.01 ^e,f,g,h,i^	52.4 ± 0.1 ^b^
F_08	3.21 ± 0.06 ^f^	0.10 ± 0.01 ^a,b^	0.09 ± 0.01 ^a,b,c,d^	20 ± 4 ^c,d^	nd	5.98 ± 0.02 ^k^	0.79 ± 0.01 ^i^	0.02 ± 0.00 ^a^	50.4 ± 0.1 ^a^
F_09	2.89 ± 0.04 ^a^	0.18 ± 0.06 ^c,d,e,f,g,h^	0.10 ± 0.01 ^d,e,f^	10 ± 1 ^b^	trace	5.00 ± 0.01 ^d,e^	0.65 ± 0.02 ^a,b^	0.03 ± 0.00 ^b,c,d^	58.0 ± 0.1 ^h^
F_10	3.12 ± 0.07 ^b,c,d,e,f^	0.23 ± 0.01 ^h,i^	0.10 ± 0.02 ^f^	30 ± 3 ^e^	nd	4.80 ± 0.01 ^c^	0.57 ± 0.03 ^d,e,f^	0.04 ± 0.00 ^g,h,i^	59.0 ± 0.1 ^i,j^
F_11	2.98 ± 0.03 ^a,b,c,d^	0.20 ± 0.09 ^f,g,h^	0.09 ± 0.01 ^a,b,c,d,e^	5 ± 2 ^a^	nd	5.30 ± 0.01 ^g^	0.64 ± 0.02 ^g^	0.04 ± 0.00 ^e,f,g,h,i^	54.1 ± 0.1 ^d^
F_12	3.12 ± 0.05 ^b,c,d,e,f^	0.23 ± 0.01 ^h,i^	0.10 ± 0.01 ^d,e,f^	nd	nd	5.20 ± 0.01 ^f^	0.72 ± 0.02 ^c,d,e^	0.03 ± 0.00 ^b,c,d^	55.4 ± 0.1 ^e,f^
F_13	3.21 ± 0.05 ^f^	0.21 ± 0.02 ^g,h^	0.11 ± 0.01 ^f^	5 ± 1 ^a^	nd	4.80 ± 0.01 ^c^	0.70 ± 0.02 ^b^	0.03 ± 0.00 ^c,d,e,f,g,h^	56.0 ± 0.1 ^f^
F_14	3.13 ± 0.07 ^c,d,e,f^	0.19 ± 0.01 ^e,f,g,h^	0.10 ± 0.01 ^d,e,f^	nd	nd	5.00 ± 0.01 ^d,e^	0.73 ± 0.03 ^c,d^	0.03 ± 0.01 ^c,d,e,f,g^	53.0 ± 0.1 ^b,c^
F_15	2.91 ± 0.02 ^a,b^	0.14 ± 0.03 ^a,b,c,d,e^	0.11 ± 0.01 ^e,f^	nd	nd	5.00 ± 0.01 ^d,e^	0.93 ± 0.02 ^f,g^	0.05 ± 0.00 ^i^	56.0 ± 0.1 ^f^
F_16	2.89 ± 0.05 ^a^	0.14 ± 0.01 ^b,c,d,e,f,g^	0.11 ± 0.01 ^b,c,d,e^	20 ± 3 ^c^	trace	5.05 ± 0.01 ^e^	0.56 ± 0.02 ^d,e,f^	0.04 ± 0.00 ^e,f,g,h,i^	55.0 ± 0.1 ^e^
F_17	2.97 ± 0.03 ^a,b,c,d^	0.14 ± 0.00 ^a,b,c,d,e,f^	0.10 ± 0.01 ^a,b,c,d,e^	32 ± 1 ^e^	nd	5.60 ± 0.01 ^i^	0.75 ± 0.03 ^f,g^	0.04 ± 0.00 ^d,e,f,g,h,i^	53.0 ± 0.1 ^b,c^
F_18	2.98 ± 0.04 ^a,b,c,d,e^	0.08 ± 0.01 ^a^	0.10 ± 0.00 ^a,b,c,d,e^	nd	nd	5.78 ± 0.01 ^j^	0.71 ± 0.02 ^e,f^	0.04 ± 0.01 ^e,f,g,h,i^	50.0 ± 0.1 ^a^
F_19	3.08 ± 0.08 ^a,b,c,d,e,f^	0.23 ± 0.05 ^h,i^	0.10 ± 0.01 ^a,b,c,d,e^	25 ± 2 ^d^	nd	4.90 ± 0.01 ^d^	0.79 ± 0.01 ^g^	0.04 ± 0.00 ^h,i^	57.0 ± 0.1 ^g^
F_20	3.11 ± 0.10 ^b,c,d,e,f^	0.10 ± 0.02 ^a,b^	0.11 ± 0.01 ^b,c,d,e^	30 ± 3 ^e^	trace	5.00 ± 0.01 ^d,e^	0.51 ± 0.02 ^a,b^	0.03 ± 0.00 ^c,d,e,f^	56.0 ± 0.1 ^f^
F_21	3.09 ± 0.10 ^a,b,c,d,e,f^	0.14 ± 0.02 ^a,b,c,d,e,f^	0.11 ± 0.01 ^b,c,d,e^	20 ± 1 ^c^	nd	4.80 ± 0.01 ^c^	0.54 ± 0.01 ^a,b^	0.03 ± 0.00 ^c,d,e,f,g,h^	57.0 ± 0.1 ^g^
F_22	2.96 ± 0.09 ^a,b,c,d^	0.14 ± 0.01 ^a,b,c,d,e^	0.11 ± 0.01 ^c,d,e,f^	nd	nd	4.60 ± 0.01 ^a^	0.68 ± 0.01 ^c,d,e,f^	0.03 ± 0.00 ^b,c^	60.0 ± 0.1 ^k^

nd—not detected. Different lower-case letters in the same column indicate significantly different values (*p* ˂ 0.05), according to post hoc Tukey’s HSD test.

**Table 3 foods-12-02291-t003:** Waxes (W), moisture (M), free fatty acids (FFA), phospholipids (P), soap (S), total carotenoids content (C), iron (Fe), copper (Cu), as well as oil transparency (T) filtration removal efficiency assisted by cellulose-based filtration aids.

Filtration Cycle	Parameter								
	W (mg kg^−1^)	M (%)	FFA (%)	P (mg kg^−1^)	S (mg kg^−1^)	C (mg kg^−1^)	Fe (mg kg^−1^)	Cu [mg kg^−1^)	T [%)
F_01	−99.23 ± 0.02 ^f,g^	−10.09 ± 4.87 ^a,b^	2.27 ± 21.68 ^a,b,c,d,e^	−54.28 ± 3.68 ^g,h^	−100.00 ± 0.00 ^a^	−6.70 ± 0.56 ^g,h^	−82.01 ± 2.92 ^a^	−41.08 ± 2.32 ^b^	7.00 ± 0.11 ^g,h,i^
F_02	−99.18 ± 0.03 ^g,h^	15.26 ± 12.60 ^b^	−9.09 ± 9.09 ^a,b,c^	−69.40 ± 9.53 ^e,f^	−100.00 ± 0.00 ^a^	−4.63 ± 0.12 ^i,j^	−5.52 ± 8.74 ^d,e,f^	−20.33 ± 10.88 ^b,c,d^	5.35 ± 0.10 ^c,d,e,f^
F_03	−99.37 ± 0.01 ^a,b,c^	2.46 ± 13.15 ^a,b^	11.57 ± 22.24 ^a,b,c,d,e,f^	−82.95 ± 2.77 ^c,d^	−100.00 ± 0.00 ^a^	−5.85 ± 1.35 ^h,i^	−19.50 ± 3.36 ^b,c,d^	−4.20 ± 19.29 ^c,d^	3.90 ± 0.92 ^b,c^
F_04	−99.42 ± 0.02 ^a^	143.18 ± 10.69 ^e^	0.00 ± 0.00 ^a,b,c,d,e^	−52.28 ± 5.04 ^g,h^	−100.00 ± 0.00 ^a^	−2.49 ± 0.20 ^k^	−6.41 ± 8.23 ^d,e,f^	−0.75 ± 7.73 ^d^	2.41 ± 0.17 ^a,b^
F_05	−99.26 ± 0.05 ^e,f^	30.28 ± 4.14 ^b,c^	−19.09 ± 8.67 ^a^	0.00 ± 0.00 ^i^	−100.00 ± 0.00 ^a^	−3.86 ± 0.75 ^j,k^	−3.42 ± 5.28 ^e,f^	−29.77 ± 13.93 ^b,c,d^	4.99 ± 0.18 ^c,d,e,f^
F_06	−99.09 ± 0.02 ^i^	−53.56 ± 3.30 ^a^	−10.74 ± 0.64 ^a,b,c^	−100.00 ± 0.00 ^a^	−100.00 ± 0.00 ^a^	−15.52 ± 0.99 ^a^	−1.64 ± 0.40 ^f^	−14.57 ± 2.93 ^b,c,d^	13.44 ± 0.26 ^l^
F_07	−99.00 ± 0.02 ^j^	−20.32 ± 6.20 ^a,b^	−15.45 ± 10.24 ^a,b^	−100.00 ± 0.00 ^a^	−100.00 ± 0.00 ^a^	−10.99 ± 0.20 ^c,d^	−11.84 ± 7.52 ^b,c,d,e,f^	−12.62 ± 14.10 ^b,c,d^	9.08 ± 0.13 ^j,k^
F_08	−99.42 ± 0.01 ^a^	−6.36 ± 5.53 ^a,b^	−9.39 ± 9.11 ^a,b,c^	−74.50 ± 5.04 ^d,e^	−100.00 ± 0.00 ^a^	−8.60 ± 0.72 ^e,f^	−10.13 ± 8.78 ^c,d,e,f^	−2.03 ± 21.84 ^d^	5.61 ± 0.24 ^d,e,f,g^
F_09	−99.00 ± 0.01 ^j^	−16.52 ± 50.32 ^a,b^	9.39 ± 0.52 ^a,b,c,d,e,f^	−74.93 ± 2.59 ^d,e^	−100.00 ± 0.00 ^a^	−7.86 ± 1.11 ^f,g^	−19.30 ± 1.38 ^b,c,d^	−93.95 ± 0.80 ^a^	7.61 ± 0.17 ^i,j^
F_10	−99.41 ± 0.02 ^a^	−13.72 ± 7.65 ^a,b^	17.73 ± 7.51 ^b,c,d,e,f^	−60.53 ± 1.15 ^f,g^	−100.00 ± 0.00 ^a^	−13.46 ± 0.47 ^b^	−2.40 ± 4.52 ^e,f^	−90.48 ± 0.41 ^a^	4.11 ± 0.57 ^c,d^
F_11	−99.16 ± 0.02 ^g,h,i^	−23.57 ± 20.79 ^a,b^	0.00 ± 10.00 ^a,b,c,d,e^	−87.22 ± 3.13 ^b,c^	−100.00 ± 0.00 ^a^	−9.86 ± 0.30 ^d,e^	−6.96 ± 6.18 ^d,e,f^	−91.06 ± 0.24 ^a^	3.91 ± 0.42 ^b,c^
F_12	−99.36 ± 0.02 ^a,b,c,d^	−16.04 ± 13.17 ^a,b^	6.36 ± 5.53 ^a,b,c,d,e,f^	−100.00 ± 0.00 ^a^	−100.00 ± 0.00 ^a^	−10.29 ± 0.50 ^d,e^	−11.07 ± 0.20 ^c,d,e,f^	−25.19 ± 1.05 ^b,c,d^	4.59 ± 0.30 ^c,d,e^
F_13	−99.38 ± 0.01 ^a,b^	−12.47 ± 2.93 ^a,b^	18.28 ± 1.67 ^b,c,d,e,f^	−93.73 ± 1.41 ^a,b^	−100.00 ± 0.00 ^a^	−12.30 ± 0.45 ^b,c^	−11.47 ± 7.51 ^b,c,d,e,f^	−28.98 ± 4.62 ^b,c,d^	1.76 ± 0.38 ^a^
F_14	−99.37 ± 0.01 ^a,b,c,d^	−28.84 ± 10.29 ^a,b^	0.25 ± 8.71 ^a,b,c,d,e^	−100.00 ± 0.00 ^a^	−100.00 ± 0.00 ^a^	−9.74 ± 0.77 ^d,e^	−10.06 ± 2.29 ^c,d,e,f^	−22.50 ± 16.82 ^b,c,d^	3.99 ± 0.82 ^c^
F_15	−98.98 ± 0.01 ^j^	−6.29 ± 29.19 ^a,b^	5.81 ± 5.04 ^a,b,c,d,e,f^	−100.00 ± 0.00 ^a^	−100.00 ± 0.00 ^a^	−13.74 ± 0.20 ^a,b^	−20.00 ± 0.22 ^b,c,d^	−6.49 ± 6.61 ^c,d^	5.66 ± 0.50 ^d,e,f,g^
F_16	−98.97 ± 0.03 ^j^	30.56 ± 39.38 ^b,c^	23.61 ± 13.25 ^c,d,e,f,g^	−60.04 ± 3.20 ^f,g^	−100.00 ± 0.00 ^a^	−5.49 ± 0.21 ^h,i,j^	−23.61 ± 0.77 ^b,c^	−22.94 ± 11.58 ^b,c,d^	7.49 ± 0.49 ^h,i^
F_17	−99.13 ± 0.01 ^h,i^	−28.23 ± 18.86 ^a,b^	19.44 ± 7.35 ^b,c,d,e,f,g^	−46.69 ± 2.45 ^h^	−100.00 ± 0.00 ^a^	−4.60 ± 0.17 ^i,j^	−16.47 ± 1.68 ^b,c,d,e^	−7.54 ± 2.44 ^c,d^	6.35 ± 1.11 ^f,g,h,i^
F_18	−99.15 ± 0.02 ^h,i^	6.67 ± 30.55 ^a,b^	−4.88 ± 14.36 ^a,b,c,d^	−100.00 ± 0.00 ^a^	−100.00 ± 0.00 ^a^	−6.27 ± 0.18 ^g,h,i^	−19.65 ± 4.55 ^b,c,d^	−7.51 ± 5.48 ^c,d^	6.38 ± 0.43 ^f,g,h,i^
F_19	−99.31 ± 0.01 ^c,d,e^	160.13 ± 16.37 ^e^	55.56 ± 11.98 ^g^	−61.55 ± 0.95 ^f,g^	−100.00 ± 0.00 ^a^	−4.56 ± 0.22 ^i,j^	−9.31 ± 1.02 ^c,d,e,f^	−7.49 ± 3.33 ^c,d^	7.55 ± 0.68 ^h,i,j^
F_20	−99.33 ± 0.03 ^b,c,d^	−46.67 ± 5.77 ^a^	34.79 ± 19.41 ^e,f,g^	−59.54 ± 1.27 ^g^	−100.00 ± 0.00 ^a^	−2.35 ± 0.20 ^k^	−25.85 ± 3.93 ^b^	−32.97 ± 2.04 ^b,c^	5.98 ± 0.77 ^e,f,g,h^
F_21	−99.30 ± 0.04 ^d,e,f^	82.14 ± 15.57 ^c,d^	39.29 ± 3.09 ^f,g^	−71.60 ± 2.24 ^e^	−100.00 ± 0.00 ^a^	6.90 ± 0.01 ^m^	−23.42 ± 0.22 ^b,c^	−23.49 ± 1.40 ^b,c,d^	5.86 ± 0.25 ^e,f,g^
F_22	−99.10 ± 0.03 ^i^	142.22 ± 15.40 ^d,e^	27.78 ± 20.03 ^d,e,f,g^	−100.00 ± 0.00 ^a^	−100.00 ± 0.00 ^a^	3.29 ± 0.57 ^l^	−16.78 ± 0.28 ^b,c,d,e^	−29.37 ± 4.96 ^b,c,d^	9.58 ± 0.31 ^k^
Average	−99.22 ± 0.15	15.01 ± 61.33	9.25 ± 18.93	−74.97 ± 25.08	−100.00 ± 0.00	−6.77 ± 5.37	−16.22 ± 16.37	−27.97 ± 28.20	6.03 ± 2.56

Different lower-case letters in the same column indicate significantly different values (*p* ˂ 0.05), according to post hoc Tukey’s HSD test.

**Table 4 foods-12-02291-t004:** Artificial neural network model summary (performance and errors).

Network Name	Performance Train	Error Train	Training Algorithm	Error Function	Hidden Activation	Output Activation
MLP 2-9-9	1.000	0.455	BFGS 153	SOS	Logistic	Logistic

Performance term represents the coefficients of determination, while error terms indicate a lack of data for the ANN model.

## Data Availability

Data presented in this study are available upon request from the corresponding author. The data is not publicly available due to obligations related to the items listed in the Funding and Acknowledgments sections.

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
