# Peer review of "Sunflower Oil Winterization Using the Cellulose-Based Filtration Aid—Investigation of Oil Quality during Industrial Filtration Probe"

_foods, 2023, doi:10.3390/foods12122291_

Round 1

Reviewer 1 Report

Comments and Suggestions for Authors

The manuscript entitled “Sunflower Oil Winterization Using the Cellulose-Based Filtration Aid – Investigation of Oil Quality during Industrial Filtration Probe “has been investigated in detail. The topic addressed in the manuscript is really interesting and the manuscript contains some practical meanings. However, there are some issues which should be addressed by the authors in detail:

·         The literature review of the paper is weak! Please add several related and new references to the text. There is no 2023 citation in the paper and there is only one 2022 citation! Please update the references list.

·         I recommend the authors to review other recently developed works.

·         How the best number of epochs was obtained? Please add to the text.

·         Why only one hidden layer was used for model? Whether different architectures have been tested or not?

·         The authors claims that optimal ANN has been modeled. How was the optimized architecture (number of neurons, epochs, kind of activation function and etc.) of network obtained? Please discuss it in the manuscript precisely.

·         Why do the authors use only MLP network? The authors could compare the mentioned method with other efficient and updated methods.

·         Why the overfitting problem has not been occurred for different models?

·         It will be helpful to the readers if some discussions about insight of the main results are added as Remarks.

·         The output activation in Table 4 is “logsig”, is it true? I think it should be “purelin”, please check it.

This study may be consider for publication if it is addressed in the specified problems.

Comments on the Quality of English Language

Minor editing of English language required.

Author Response

Reviewer 1

The manuscript entitled “Sunflower Oil Winterization Using the Cellulose-Based Filtration Aid – Investigation of Oil Quality during Industrial Filtration Probe “has been investigated in detail. The topic addressed in the manuscript is really interesting and the manuscript contains some practical meanings. However, there are some issues which should be addressed by the authors in detail:

AUTHORS: The authors would like to thank the Reviewer on generous comments on our study and effort to improve the paper. The authors also gave effort to provide answers to all new questions and comments of the Reviewer and hope that answers will be satisfactory.

  • The literature review of the paper is weak! Please add several related and new references to the text. There is no 2023 citation in the paper and there is only one 2022 citation! Please update the references list.
  • I recommend the authors to review other recently developed works.

AUTHORS: The authors would like to thank the Reviewer for bringing the references to our attention. The paper is supplemented with the latest references:

  1. Guo, Y., Jia, Z., Wan, L., Cao, J., Fang, Y., Zhang, W. Effects of refining process on Camellia vietnamensis oil: Phytochemical composition, antioxidant capacity, and anti-inflammatory activity in THP-1 macrophages. Food Biosci. 2023, 52, 102440.
  2. Wen, C., Shen, M., Liu, G., Liu, X., Liang, L., Li, Y., Zhang, Y., Xu, X. (2023). Edible vegetable oils from oil crops: Preparation, refining, authenticity identification and application. Process Biochem. 2023, 124, 168-179.
  3. Chew, S.C., Ali, M.A. Recent advances in ultrasound technology applications of vegetable oil refining. Trends Food Sci Technol 2021, 116, 468-479.
  4. Ye, Z., Liu, Y. Polyphenolic compounds from rapeseeds (Brassica napus): The major types, biofunctional roles, bioavailability, and the influences of rapeseed oil processing technologies on the content. Food Res. Int. 2023, 163, 112282.
  5. Ma, G., Wang , Y., Li , Y., Zhang , L., Gao, Y., Li , Q., Yu, X. Antioxidant properties of lipid concomitants in edible oils: A review. Food Chem. 2023, 422, 136219.
  6. Ma, H., Ding, F., Wang, Y. A novel multi-innovation gradient support vector machine regression method. ISA Trans. Press, 2022, 130, 343–359.
  7. Wang, C. Peng, G. De Baets B. Embedding metric learning into an extreme learning machine for scene recognition. Expert Syst. Appl. 2022, 203, 117505.
  8. Su, J., Wang, Y., Niu, X., Shaa, S., Yu J. Prediction of ground surface settlement by shield tunneling using XGBoost and Bayesian optimization. Appl. Artif. Intel. 2022, 114, 105020.
  9. Mahmood, J., Mustafa, G. e., Ali, M. Accurate estimation of tool wear levels during milling, drilling and turning operations by designing novel hyperparameter tuned models based on LightGBM and stacking. Measurement 2022, 190, 110722.
  10. Dutta, J.; Roy, S. Occupancy Sense: context-based indoor occupancy detection & prediction using CatBoost model. Soft Comput. 2022, 119, 108536.
  • How the best number of epochs was obtained? Please add to the text.

AUTHORS: Thank you very much for this comment. As mentioned in the Manuscript, the experimentally obtained database which covered measured input and output parameters was transformed using min-max normalization scheme prior the calculation, This database was randomly divided into training, cross-validation, and testing groups (60%, 20%, and 20%, respectively). Throughout the learning procedure, ANN inputs were supplied with training set of parameters, in order to establish the optimal number of neurons in the hidden layer, to estimate the weights and bias coefficients and non-linear activation functions for every neuron in the ANN model.

During the iterative process of calculating weights and biases coefficients and testing different activation functions for the hidden and output layers, the Broyden-Fletcher-Goldfarb-Shanno (BFGS) algorithm was employed. Various activation functions were explored, including hyperbolic tangent, logistic sigmoidal, exponential, and identity functions. The identity function directly passes the activation level from the input as the output of the neurons. Logistic sigmoid uses the S-shaped logistic sigmoid function, producing output values in the range of 0 to +1. The hyperbolic tangent function (tanh) is another symmetric S-shaped (sigmoid) function, with output values ranging from -1 to +1. It often outperforms the logistic sigmoid function due to its symmetry. The exponential function utilizes the negative exponential activation function [56].The Broyden-Fletcher-Goldfarb-Shanno (BFGS) algorithm was employed during the iterative process of weights and biases coefficients calculation [56]. A sequence of distinct MLP-formed ANN layouts was investigated, altering the number of hidden neurons (between 5 to 20) introducing random initial values of weights and biases coefficients. The learning procedure of the network was repeated 100,000 times [57]. The optimization set-up included the minimization of the square error. It is assumed that the successful train-ing was reached when learning and cross-validation curves approached zero.

To investigate the removal efficiency of chemical parameters during winterization, by cooling and filtration, an Artificial Neural Network (ANN) technique was employed. The structure and outcomes of the ANN model depend on the initial assumptions of matrix parameters, which are crucial for building and fitting the ANN to experimental data. Also, the behavior of the ANN model can be influenced by the number of neurons in the hidden layer. To address this concern, each network topology was iterated 100,000 times to minimize random correlations caused by initial assumptions and random weight initialization. Through this approach, the highest r2 value during the training cycle was achieved when using nine hidden neurons for constructing the ANN model (Figure 3a).

The model underwent training for 100 epochs, and Figure 3b displays the training results, specifically the train accuracy and error (loss). During the training process, the training accuracy consistently improved as the number of training cycles increased, up until the 70th to 80th epoch. At this point, the training accuracy reached a nearly constant value. The 70th to 80th epoch yielded the highest train accuracy and lowest train loss. However, after this point, a slight increase in train accuracy and decrease in train loss were observed, indicating the onset of overfitting. Going beyond 80 epochs for training could potentially lead to significant overfitting, while training for 70 epochs would be sufficient to achieve high model accuracy without risking overfitting (Figure 3b).

  • Why only one hidden layer was used for model? Whether different architectures have been tested or not?

The artificial neural networks (ANNs) are one of the most powerful computer modeling techniques, based on statistical approach, currently being used in many fields of engineering to simulate the complex relationships which are difficult to describe with physical models [Taylor, 2000]. In this article, a multi-layer perceptron model (MLP) that consists of three layers (input, hidden and output) was evaluated. This architecture is called feed forward because inputs propagate through layers in a forward progression. Such a model is the most common, flexible and general-purpose kind of ANN. These architectures are used in prediction, and have been proven quite capable of approximating nonlinear functions [Taylor, 2000], giving the reason for choosing it in this study.

A trial and error procedure is necessary to perform before modeling, until a good network behavior was obtained, as well as to choose the number of hidden layers, and the number of processing elements (also called “neurons”) in hidden layer(s). The use of just one layer is advisable, because more layers exacerbates the problem of local minima. The network has been trained with Levenberg–Marquardt algorithm due to its high accuracy in similar function approximation [Taylor, 2000]. Numbers of neurons in the input and the output layers are determined by the number of corresponding variables, respectively. In order to find an optimal architecture, different number of neurons in hidden layer was taken into consideration [Taylor, 2000] and the sum of squares error for each network was calculated, as StatSoft Statistica's default. In this study, the number of hidden neurons varied from 5–20 in developed networks, with 2 inputs and 9 outputs, bound with 86–249 weight coefficients depending on the number of hidden neurons.

A neural network tries to find a function which, when given the inputs, produces the outputs. The information passes between layers through a transfer or “activation” function is a typically nonlinear function for hidden layers and linear for the output layer. Most common nonlinear activation functions are logistic sigmoid and hyperbolic tangent functions (also exponential, sine, softmax, Gausian etc.). In most applications, hyperbolic tangent function behaved better compared to other functions [Taylor, 2000].

Reference:

Taylor, B.J., Methods and Procedures for the Verification and Validation of Artificial Neural

Networks. Springer Science+Business Media, Inc., USA, 2000.

  • The authors claims that optimal ANN has been modeled. How was the optimized architecture (number of neurons, epochs, kind of activation function and etc.) of network obtained? Please discuss it in the manuscript precisely.

AUTHORS: As mentioned in the Manuscript, the experimentally obtained database which covered measured input and output parameters was transformed using min-max normalization scheme prior the calculation, This database was randomly divided into training, cross-validation, and testing groups (60%, 20%, and 20%, respectively). Throughout the learning procedure, ANN inputs were supplied with training set of parameters, in order to establish the optimal number of neurons in the hidden layer, to estimate the weights and bias coefficients and non-linear activation functions for every neuron in the ANN model.

During the iterative process of calculating weights and biases coefficients and testing different activation functions for the hidden and output layers, the Broyden-Fletcher-Goldfarb-Shanno (BFGS) algorithm was employed. Various activation functions were explored, including hyperbolic tangent, logistic sigmoidal, exponential, and identity functions. The identity function directly passes the activation level from the input as the output of the neurons. Logistic sigmoid uses the S-shaped logistic sigmoid function, producing output values in the range of 0 to +1. The hyperbolic tangent function (tanh) is another symmetric S-shaped (sigmoid) function, with output values ranging from -1 to +1. It often outperforms the logistic sigmoid function due to its symmetry. The exponential function utilizes the negative exponential activation function [56].The Broyden-Fletcher-Goldfarb-Shanno (BFGS) algorithm was employed during the iterative process of weights and biases coefficients calculation [56]. A sequence of distinct MLP-formed ANN layouts was investigated, altering the number of hidden neurons (between 5 to 20) introducing random initial values of weights and biases coefficients. The learning procedure of the network was repeated 100,000 times [57]. The optimization set-up included the minimization of the square error. It is assumed that the successful train-ing was reached when learning and cross-validation curves approached zero.

  • Why do the authors use only MLP network? The authors could compare the mentioned method with other efficient and updated methods.

AUTHORS: Various classical machine learning models are widely utilized in modeling across different scientific fields. These models include artificial neural network (ANN), random forest regression (RFR), support vector machine (SVM), extreme learning machine (ELM), K-nearest neighbors (KNN), and decision tree (DT). SVM, a discriminant technique based on statistical learning theory, is recognized for its exceptional generalization ability. By striking a balance between model complexity and training error, the optimal network is achieved [46]. ELM constructs a single-layer feed-forward network by randomly generating input weights and biases for the hidden layers [47].

State-of-the-art machine learning techniques offer a diverse range of options for se-quence data, such as ensemble learning models like XGBoost [48], LightGBM [49], and CatBoost. XGBoost stands out for its high prediction accuracy and interpretability. LightGBM allows efficient handling of large datasets and GPU training. Compared to XGBoost, LightGBM models have demonstrated superior accuracy and faster performance. Data fusion, incorporating gradient boosting with categorical attributes supported by the CatBoost algorithm, enhances forecasting accuracy [50].

In this paper, artificial neural network model [51,52], as well known and broadly accepted machine learning technique was utilized to contemplate the removal efficiency of chemical parameters (the contents of: wax, moisture, phospholipids, soap, fatty acids and carotenoids), oil transparency (at wavelength of 455 nm), Fe and Cu content after filtration process (Table 3), based on data of initial values of chemical parameters, Fe and Cu content, and also the data regarding filtration time and the quantity of filtration agent (Table 1).

References:

  1. Ma, H., Ding, F., Wang, Y. A novel multi-innovation gradient support vector machine regression method. ISA Trans. Press, 2022, 130, 343–359.
  2. Wang, C. Peng, G. De Baets B. Embedding metric learning into an extreme learning machine for scene recognition. Expert Syst. Appl. 2022, 203, 117505.
  3. Su, J., Wang, Y., Niu, X., Shaa, S., Yu J. Prediction of ground surface settlement by shield tunneling using XGBoost and Bayesian optimization. Appl. Artif. Intel. 2022, 114, 105020.
  4. Mahmood, J., Mustafa, G. e., Ali, M. Accurate estimation of tool wear levels during milling, drilling and turning operations by designing novel hyperparameter tuned models based on LightGBM and stacking. Measurement 2022, 190, 110722.
  5. Dutta, J.; Roy, S. Occupancy Sense: context-based indoor occupancy detection & prediction using CatBoost model. Soft Comput. 2022, 119, 108536.
  • Why the overfitting problem has not been occurred for different models?

AUTHORS: The model underwent training for 100 epochs, and Figure 3b displays the training results, specifically the train accuracy and error (loss). During the training process, the training accuracy consistently improved as the number of training cycles increased, up until the 70th to 80th epoch. At this point, the training accuracy reached a nearly constant value. The 70th to 80th epoch yielded the highest train accuracy and lowest train loss. However, after this point, a slight increase in train accuracy and decrease in train loss were observed, indicating the onset of overfitting. Going beyond 80 epochs for training could potentially lead to significant overfitting, while training for 70 epochs would be sufficient to achieve high model accuracy without risking overfitting (Figure 3b).

  • It will be helpful to the readers if some discussions about insight of the main results are added as Remarks.

AUTHORS: Following sentences were added in the 3. Results and Discussion section:

The investigation results of the content of: waxes, moisture, phospholipids, soap, free fatty acids, iron, copper, total carotenoids content, as well as oil transparency before and after filtration are shown in Table 1 and Table 2. The content of the mentioned parameters in the tested oil samples depends on the crude oil itself and on the previous stages of refining, while the content after filtration is affected by the filtration conditions (quantity of added filtration aid and filtration time).

  • The output activation in Table 4 is “logsig”, is it true? I think it should be “purelin”, please check it.

AUTHORS: The developed optimal neural network model showed the adequate generalization capabilities for the prediction of the removal efficiency of: chemical parameters (the contents of: wax, moisture, phospholipids, soap, fatty acids and carotenoids), oil transparency (at wavelength of 455 nm), Fe and Cu content after filtration process (Table 3), com-pared to initial, based on data of initial values of chemical parameters, Fe and Cu content, and also the data regarding filtration time and the quantity of filtration agent. The optimum number of neurons in the hidden layer of ANN model was 9 (network MLP 2-9-9), while the r2 values were equal to 1.000, during the training cycle r2 for output variables, hidden and output layers  activation functions were logistic sigmoid  (Table 4).

The developed ANN model for the prediction of the removal efficiency of chemical parameters (the contents of: wax, moisture, phospholipids, soap, fatty acids and carotenoids), oil transparency (at wavelength of 455 nm), Fe and Cu content after filtration process, consisted of 117 weights-bias coefficients due showing the high nonlinearity of the system [59,60].

Table 4. Artificial neural network model summary (performance and errors)

Network

name

Performance

Train

Error

Train

Training

algorithm

Error

function

Hidden

activation

Output

activation

MLP 2-9-9

1.000

0.455

BFGS 153

SOS

Logistic

Logistic

*Performance term represents the coefficients of determination, while error terms indicate a lack of data for the ANN model

This study may be consider for publication if it is addressed in the specified problems.

Reviewer 2 Report

Comments and Suggestions for Authors

Minor revision   this is an interesting article about investigating sunflower oil quality after cellulose-based filtration. The manuscript and the data presented are clear and consistent.  The overall language is good. Most sections are well prepared. I would be grateful if the authors could improve the manuscript by eliminating typos across the whole text.  Here are some suggestions: Abstract: please add the instruments used for the analyses. Materials and Methods: please add the botanical name of the plant, the year and the site where the oil samples were collected. Please, add the voucher specimen if possible. Conclusions: please reduce the section and add future perspectives. Bibliography: please check typos.

Author Response

Minor revision this is an interesting article about investigating sunflower oil quality after cellulose-based filtration. The manuscript and the data presented are clear and consistent.  The overall language is good. Most sections are well prepared.

I would be grateful if the authors could improve the manuscript by eliminating typos across the whole text. 

AUTHORS: The authors would like to thank the Reviewer on comments on our study and effort to improve the paper. The authors greatly appreciate the suggestions received and will try to adopt all of them.

The entire paper has been revised and typographical errors have been corrected. All corrections are marked in red in the manuscript.

Here are some suggestions:

Abstract: please add the instruments used for the analyses.

AUTHORS: Information about the instruments used for the analyses were added in the Abstract:

In order to investigate the mentioned parameters following techniques were used: gravimetric (waxes and moisture content), spectrophotometric (phospholipids and carotenoids content, oil transparency), volumetric (soaps and free fatty acids content) as well as inductively coupled plasma mass spectrometry (ICP-MS) for Fe and Cu content.

Materials and Methods: please add the botanical name of the plant, the year and the site where the oil samples were collected. Please, add the voucher specimen if possible.

AUTHORS: Since experiment was carried out in the industrial conditions, as a part of regular industrial production, with high capacity (industrial refining capacity was 200 t of crude oil/day), voucher specimen is not eligible.

The 2.1. Materials and Methods section (2.1. Samples section) was supplemented with the required information:

Industrial processing was carried out during 2019-2020, the industrial refining capacity was 200 tons of crude sunflower oil per day, every 24 hours. The obtained oil is produced from sunflower seeds (Helianthus annuus L.) grown in the territory of Vojvodina (north of the Republic of Serbia) in 2019.

Conclusions: please reduce the section and add future perspectives.

AUTHORS: According to the reviewer's request, the 4. Conclusions section was reduced, and future research plans were added. These sentences were added in the paper:

Based on the obtained results, it was concluded that there is a possibility to change the quantity of added filtration aid and to achieve the desired result. Therefore, future plans are to carry out research on the optimization of the filtration aids quantity and filtration time aimed to reduce the operational costs.

Bibliography: please check typos.

AUTHORS: The entire paper, as well as References section, has been revised and typographical errors have been corrected.

Reviewer 3 Report

Comments and Suggestions for Authors

1. The sunflower oil properties before and after treatment consisting of winterization and filtration using a horizontal pressure leaf filter together with cellulose-based filtration agents were experimentally investigated in this study. The artificial neural network (ANN) model was also used to evaluate the effects of either one parameter on the other remaining parameters. The properties of wax, moisture, phospholipids, soaps, fatty acids, oil transparency, carotenoids, and Fe and Cu contents were measured and compared.

2.  The research gaps to break through in this study and the novelty of this study could be further emphasized and clarified at the end of Section Introduction. In addition, the results of this study could be explained and discussed much more profoundly in Section 3. (Results and Discussion). Moreover, the manuscript could be further carefully checked its writing since a few typos and grammatical errors are easily found.

3. The manuscript could be checked thoroughly, for example, repeated content in lines 205-209 with that in lines 214-218; and the content in lines 398-401 is repeated with that in lines 403-405; typo errors “sand” in line 399, r2 in lines 426 and 429; and grammatical errors “Prior” in line 224, repeated words in line 285, and please rephrase the sentence in line 290 and the expression in line 304.

4. The result variations of those parameters such as W, M, FFA, and P in Tables 1-3, respectively among various filtration cycles from F_01 to F_22 could be further explained and discussed.

5. In Table 3, for a decimal number such as 99.23, decimal point “.” instead of current comma “,” symbol should be used.

6. A flow chart that includes the winterization, filtration, and other processes for this study is suggested to be supplemented to make the explanation for the operation procedures more illustrative.

7. The characters and numbers in Figure 2 are too small. This figure image could also be sharper.

8. Whole words could be provided once their corresponding abbreviation first appears, for example, SOS in line 407 and RFR in line 428.

Comments on the Quality of English Language

1. The whole manuscript writing is suggested to be carefully checked by experts. Typo and grammatical errors could be easily found. A few sentences could be rephrased to make those meanings more understandable. 

Author Response

Comments and Suggestions for Authors

  1. The sunflower oil properties before and after treatment consisting of winterization and filtration using a horizontal pressure leaf filter together with cellulose-based filtration agents were experimentally investigated in this study. The artificial neural network (ANN) model was also used to evaluate the effects of either one parameter on the other remaining parameters. The properties of wax, moisture, phospholipids, soaps, fatty acids, oil transparency, carotenoids, and Fe and Cu contents were measured and compared.

AUTHORS: The authors would like to thank the Reviewer on comments on the manuscript content.

  1. The research gaps to break through in this study and the novelty of this study could be further emphasized and clarified at the end of Section Introduction.

AUTHORS: The authors agree with the Reviewer about the importance of explanation of the novelty of the study. Following sentences were added at the end of 1. Introduction section:

The importance of this research is in the introduction of cellulose-based filtration aids with certain technological and ecological advantages compared to classic aids used in the oil refining process. On the other hand, since their price is higher compared to classic filtration aids, it is necessary to prove the effectiveness of these filtration aids, so companies would be ready to invest. In this regard, the main objective of this work is to prove the effectiveness of cellulose-based filtration aids and to optimize the process.

In addition, the results of this study could be explained and discussed much more profoundly in Section 3. (Results and Discussion).

AUTHORS: Authors are grateful to the reviewer for his comments, but they believe that due to the large number of samples (44) and the large number of parameters (11), the paper is extensive and that a more detailed discussion of the results would overload and exceed the scope of the original scientific paper. In addition, this paper examines industrial trials, not a classically designed laboratory experiment, and the discussion of the results is in accordance with that. If the reviewer still believes that a more detailed discussion of the results is needed, the authors would ask him to get specifics on what to pay attention to, since with this comment we should change the entire 3. Results and Discussion section.

Moreover, the manuscript could be further carefully checked its writing since a few typos and grammatical errors are easily found.

AUTHORS: According to the reviewer's request the entire paper has been revised and typographical errors have been corrected. All corrections are marked in red in the manuscript.

  1. The manuscript could be checked thoroughly, for example, repeated content in lines 205-209 with that in lines 214-218; and the content in lines 398-401 is repeated with that in lines 403-405; typo errors “sand” in line 399, r2 in lines 426 and 429; and grammatical errors “Prior” in line 224, repeated words in line 285, and please rephrase the sentence in line 290 and the expression in line 304.

AUTHORS: Authors adopted all reviewers’ requests, repeated content in lines 214-218 and 403-405 in prior version of the manuscript were excluded; line 399 was excluded as a part of repeated content; r2 in lines 426 and 429 is excluded as a part of another reviewers’ request; Prior was replaced with before; lines 285, 290 and 304 were rephrased as following:

Correlation investigation found that the waxes removal efficiency on the horizontal pressure leaf filters assisted by the cellulose based aids was higher, as higher the waxes content in the oils before filtration was (R=0.98; p=0.00).

Total carotenoids (-9.69%) and moisture (-7.81%) content in the initial oil negatively influenced the wax removal.

Similar results were reported in enzymatic degummed oil (between 63.5 and 65.25 mg/kg) [66], while significantly higher total phospholipids content values in degummed sunflower oil (between 470 and 1230 mg/kg) found Lamas et al., 2016 [64].

  1. The result variations of those parameters such as W, M, FFA, and P in Tables 1-3, respectively among various filtration cycles from F_01 to F_22 could be further explained and discussed.

AUTHORS: Authors are grateful to the reviewer for his comments, but they believe that due to the large number of samples and the large number of parameters, the paper is extensive and that a more detailed discussion of the results would overload and exceed the scope of the original scientific paper. In addition, this paper examines industrial trials, not a classically designed laboratory experiment, and the discussion of the results is in accordance with that. If the reviewer still believes that a more detailed discussion of the results is needed, the authors would ask him to get specifics on what to pay attention to, since with this comment we should change the entire 3. Results and Discussion section.

  1. In Table 3, for a decimal number such as 99.23, decimal point “.” instead of current comma “,” symbol should be used.

AUTHORS: Authors apologize for making the aforementioned omission. Decimal point “.” is used in Table 3.

  1. A flow chart that includes the winterization, filtration, and other processes for this study is suggested to be supplemented to make the explanation for the operation procedures more illustrative.

AUTHORS: A flow chart (Figure 1) including winterization, filtration and other processes for this study has been drawn up and inserted in order to make the explanation of individual phases of the industrial probe of refining more illustrative.

  1. The characters and numbers in Figure 2 (in revised manuscript Figure 4) are too small. This figure image could also be sharper.

AUTHORS: The authors gave effort to increase the font size and enlarge and sharpen Figure 2 (in revised manuscript Figure 4).

  1. Whole words could be provided once their corresponding abbreviation first appears, for example, SOS in line 407 and RFR in line 428.

AUTHORS: The manuscript was modified according to the request of the reviewer (SOS and RFR were explained at their first appearance in the manuscript).

Comments on the Quality of English Language

  1. The whole manuscript writing is suggested to be carefully checked by experts. Typo and grammatical errors could be easily found. A few sentences could be rephrased to make those meanings more understandable.

AUTHORS: The entire paper has been revised by a native English speaker; all typographical and grammar errors have been corrected. All corrections are marked in red in the manuscript.

Round 2

Reviewer 1 Report

Comments and Suggestions for Authors

My recommendation is "Accept in present form".

Reviewer 3 Report

Comments and Suggestions for Authors

1. The reviewer after reading through the reply to the comments and the revised manuscript considered the manuscript has been revised based on the comments.

2. The revised manuscript is thus suggested to be acceptable in its current form.